# Tailoring Sexual Health Research Practices to Meet the Needs of Adolescent Girls in Low- and Middle-Income Countries: Findings from Mexico

Argentina E. Servin [1,*], Ruth Macklin [2], Sara Wilkerson [3], Teresita Rocha-Jiménez [4,5], Gudelia M. Rangel [6], Sophie E. O'Bryan [1] and Celia B. Fisher [3]

1   Department of Medicine, Division of Infectious Diseases and Global Public Health, University of California San Diego, 9500 Gilman Drive, MC 0507, La Jolla, CA 92093, USA; sophie.e.obryan@gmail.com
2   Department of Epidemiology and Population Health, Albert Einstein College of Medicine, 1300 Morris Park Avenue, Bronx, NY 10461, USA; ruth.macklin@einsteinmed.edu
3   Center for Ethics Education, Fordham University, Bronx, NY 10458, USA; swilkerson3@fordham.edu (S.W.); fisher@fordham.edu (C.B.F.)
4   Society and Health Research Center, Facultad de Ciencias Sociales y Artes, Universidad Mayor, Santiago 7560908, Chile; teresita.rocha@umayor.cl
5   Millennium Nucleus on Sociomedicine, Santiago 7560908, Chile
6   The U.S.-Mexico Border Health Commission, Mexico Section, Tijuana 22010, Baja California, Mexico; grangel@colef.mx
*   Correspondence: arservin@ucsd.edu; Tel.: +1-619-576-721

**Abstract:** Sexual and reproductive health (SRH) research is essential for the development of population-tailored evidence-based policies and programs that support sexual health among adolescent girls. However, ethical challenges create barriers to girls' participation in low- and middle-income countries (LMICs). From February to September 2019, girls aged 16–20 ($n$ = 30) who participated in the *Jovenes Sanos* study in Tijuana, Baja California (ClinicalTrials: NCT03660514) responded to in-depth interviews (IDs) on the perceived risks and benefits of participating in studies which address gender-based violence, unintended pregnancy, and STIs. Emergent themes indicated the need to ensure that consent and incentive procedures are tailored to the developmental level of participants, while highlighting the importance of researcher–participant relationships, and demonstrating how research can serve as an opportunity to empower girls to express their sexual health medical needs. Understanding adolescent girls' voices is a critical step in ensuring that consent to participate SRH research is tailored to the developmental needs of participants, is culturally competent, and has a participant-centered approach.

**Keywords:** sexual and reproductive health; adolescent health; ethical research; Latin America; sexual health interventions

## 1. Introduction

Data on the rates of sexually transmitted infections (STIs) and unintended pregnancy in adolescents indicate that the sexual and reproductive health (SRH) needs of adolescents are inadequately met. In 2021, approximately 14% of adolescent girls and young women worldwide experienced childbirth before turning 18 [1]. Early childbearing, encompassing pregnancy and delivery during adolescence, has the potential to disrupt the otherwise healthy development of girls as they transition into adulthood, adversely affecting their education, livelihoods, and overall health [2,3]. This need remains especially urgent in low- and middle-income countries (LMICs). Approximately 21 million girls aged 15–19 years become pregnant each year in developing countries [4]. Likewise, over one million sexually transmitted infections (STIs) are contracted globally each day. According to the World Health Organization (WHO), 374 million new infections from four treatable

STIs (chlamydia, gonorrhea, syphilis, and trichomoniasis) were reported in 2020 [5]. While STIs are widespread globally, their prevalence and the associated burden are notably higher LMICs and among adolescents and young adults.

Previous research has documented an association between gender-based violence (GBV) and an increased risk of unintended pregnancy and STIs. Gender-based violence is defined by the WHO as "any act of gender-based violence that results in, or is likely to result in, physical, sexual or psychological harm or suffering to women, including threats of such acts, coercion or arbitrary deprivation of liberty, whether occurring in public or in private life" [6]. The highest incidence of GBV is generally reported during late adolescence. In a study across 81 countries, 29% of ever-partnered adolescent girls (aged 15–19) reported that they had experienced GBV [7]. Specifically, in Mexico, there are 22.8 million adolescents (10–19) (17% of the population) [8], and it has one of the highest adolescent pregnancy rates in the world outside of sub-Saharan Africa, with 58 of every 1000 girls aged 15–19 becoming pregnant each year [9]. Furthermore, half (47%) of adolescent girls' report experiencing GBV [10]. These statistics highlight the need for research focused on the SRH of adolescents in order to understand the socio-cultural, behavioral, and environmental determinants of their health needs, and to develop age-specific evidence-based interventions, policies, and programs that support SRH among adolescent girls. Adolescents make up a significant portion of the population in LMICs. Addressing their SRH needs is crucial for the overall public health of these countries. However, LMICs commonly encounter various healthcare challenges. A significant hurdle is the inequity in healthcare access. While considerable attention has been given to enhancing access through healthcare system improvements, limited studies have examined the barriers to implementing SRH interventions or included adolescent girls in this type of research [11–14].

Conducting research focused on SRH with adolescents presents various challenges, especially in an LMIC. In Mexico, in addition to the cultural and legal constraints, a variety of ethical challenges hinder SRH research for both adolescent participants and investigators. These challenges are rooted in the complexities and uncertainties surrounding research procedures including consent and assent, risks and benefits assessment, risk management, waiver of parental permission, and confidentiality, among others [15–20]. Despite these challenges, conducting SRH research among adolescents in LMICs is essential for improving public health, promoting human rights and equity, and supporting social and economic development. It plays a critical role in empowering adolescents to make informed choices about their health and well-being, ultimately benefiting both individuals and society. Thus, the present study was conducted to collect data from adolescent girls who had participated in a previous SRH intervention study known as *Jóvenes Sanos*, a randomized control trial pilot, to contribute to enhanced ethical procedures for research centered on the SRH of adolescent girls in Mexico and other LMICs.

## 2. Materials and Methods

### 2.1. Study Overview and Design

This qualitative study reports data from a subset of individuals who participated in a larger project, *Jóvenes Sanos*, a randomized control trial pilot (K23HD084756; PI: Servin) focused on adapting a clinic-based behavioral intervention (Addressing Reproductive Coercion in Clinical Settings) to reduce risk for GBV, HIV/STIs, and unintended pregnancy among adolescent girls aged 16–20 years old (*n* = 100) accessing family planning (FP) services at community health centers in Tijuana, Mexico [21–23]. Specific information regarding the parent study design, eligibility criteria, and outcome measures are available on clinicaltrails.gov (NCT03660514). Briefly, two community health centers were randomly assigned to either 3-day eight-hour training for all FP providers on how to deliver the *Jovenes Sanos* intervention or to a standard-of-care control condition. All adolescent girls aged 16–20 seeking care in these community health centers were eligible to participate. Consenting adolescent girls used a tablet to answer survey questions immediately prior to their clinic visit, a brief exit survey immediately after the clinic visit, and a follow-up survey

12–20 weeks after the baseline visit. The survey included questions on recent experiences of GBV, including reproductive coercion, physical and sexual partner violence victimization, unintended pregnancy, recognition of sexual and reproductive coercion, self-efficacy to implement harm reduction strategies, knowledge of GBV-related resources and services, use of GBV-related resources and services [21,23].

Medical record chart reviews provided additional data about GBV (e.g., intimate partner violence, reproductive coercion) and disclosure, SRH diagnoses, health care utilization, and utilization of services for women who experience violence. The clinician-delivered intervention focused on three major components: (1) universal client education and assessment regarding GBV; (2) discussion of harm reduction behaviors to reduce risk of unintended pregnancy and GBV victimization; (3) supported referrals to GBV victim services (including provision of GBV-related resources to all clients regardless of disclosure) [21]. Additionally, participants are given a palm-sized brochure about the health impact of GBV, harm reduction (e.g., intrauterine and emergency contraception, safety planning to reduce risk of GBV), and GBV resources [21–23].

## 2.2. Study Setting

Participants for *Jovenes Sanos* and this study were recruited from two publicly funded, community-based health centers in Tijuana, Baja California, Mexico. These health centers are operated by the Mexican Ministry of Health (ISESALUD). Mexico currently offers universal access to contraceptive and sexual and reproductive health care (SRH) services via community health clinics run by ISESALUD and they adhere to the Mexican Ministry of Health's contraceptive counseling guidelines [24]. The majority of the health centers are located in highly marginalized communities and are usually the first, sometimes the only, facility available when patients are seeking care for any number of health concerns including contraceptives [25]. Two community health centers located on the east side of Tijuana were selected to participate in the adaptation and pilot of *Jovenes Sanos*. Baja California (BC), Mexico (2020 pop. 1922, 523) is one of the states with the highest levels of adolescent pregnancy and GBV in the country [26,27]. It is the largest city in the state of Baja California and the 6th largest metropolitan area in the country [26].

## 2.3. Recruitment

Briefly, from July to January 2019, adolescent girls (*n* = 100) seeking FP services were identified and recruited from two community health centers in Tijuana, Mexico operated by the Mexican Ministry of Health (ISESALUD) to participate in *Jóvenes Sanos*. Local female research assistants approached adolescent girls in the waiting rooms of the two participating community health centers to ask if they were interested in hearing more about participating in a "girls health study". If interested, participants were taken to a private setting within the clinic by the research assistant where the purpose of the study was explained to them, and they were screened for inclusion (being biologically female, aged 16–20 years old, speaking Spanish or English, seeking FP services at the participating community health center, residing in Tijuana, Mexico, having no plans to move in the next 12 months). As part of informed consent, research assistants explained that participation in the study was completely voluntary and would in no way affect the care that girls received at the health center. Those who were eligible and interested provided voluntary written informed consent before participation in the study. For the present study, from February to September 2019, adolescent girls aged 16–20 (*n* = 30) who participated in the *Jóvenes Sanos* intervention (parent study) were recontacted and invited to participate in in-depth interviews (IDIs) to capture their experiences as study participants.

## 2.4. Data Collection

A total of 30 interviews were conducted in Spanish by native Spanish-speaking female research assistants (RAs) in private rooms at the participating health centers. The interviews were audio taped (identified using only a study-unique identification number) and

lasted 60–90 min. The interview protocol was informed by the WHO guidance on ethical considerations in planning and reviewing research studies on SRH among adolescents [28] as well as our previous qualitative research with vulnerable underserved populations in this setting [29]. Further, the interview followed an open-ended guide that was iteratively revised as data collection and analysis progressed. Questions elicited adolescent girls' narratives regarding ethical issues that may have impacted their decision to participate and their experience participating in the intervention study.

*2.5. Data Analysis*

IDIs were transcribed verbatim and analyzed in Spanish by a trained bilingual research team. Qualitative analysis was led by the PI in conjunction with two members of the binational research team (i.e., one from the US-based team and one from the Mexico-based team). The research team systematically read through transcripts, engaged in open line-by-line coding, and constructed a coding scheme based on the content of the transcripts which was iteratively revised until the research team reached consensus. Transcripts were coded in ATLAS.ti version 6.2 to group, label, and describe intersections between emergent themes related to the ethical considerations for conducting SRH research among adolescents, especially research focused on the intersection of GBV and HIV/STIs in this setting [30–32]. This analysis adopted deductive and inductive perspectives in which participants' language and experiences were used to identify and understand their experiences as human subjects in research centered on GBV and HIV/STI prevention [33]. Thematic saturation was reached with the 15th interview; additional interviews were conducted to ensure no new themes emerged and to illuminate the nuances of perspectives through additional quotes. Applying the final coding scheme, inter-coder reliability was assessed and inter-coder consistency greater than 80% was achieved.

*2.6. Ethical Considerations*

The study was approved by IRBs at the University of California San Diego and the Universidad Xochicalco, in Tijuana, Mexico. Given that SRH research tends to focus on sensitive topics that adolescents may not want their parents to be privy to, request for guardian permission was waived. Adolescent girls provided written informed consent prior to study participation and were compensated 15 USD for the time and travel costs associated with their participation in this sub-study. Spanish speaking female research assistants (RAs) obtained informed consent, and the principal investigator and/or project coordinator were onsite to answer questions raised by potential participants. Additionally, a Youth Advisory Board (YAB) established by the Mexican Ministry of Health (Secretaria de Salud del Estado de Baja California) was involved in the *Jóvenes Sanos* study and the present sub-study. The YAB is comprised of 50 adolescent girls (aged 15–18) from the school districts neighboring the community health centers striving to vocalize their needs by engaging in advocacy and other activities. The YAB provided guidance in the development of the intervention materials for *Jóvenes Sanos*, the consent forms, and the IDI guidelines developed for the present study. For ethical and confidentiality purposes, names of participants have been changed.

## 3. Results
### *3.1. Participant Characteristics*

Table 1 summarizes the sociodemographic characteristics of the 30 participants. The mean age was 17.6 and only 20% (*n* = 6) were currently enrolled in school. All participants (*n* = 30) were of Mexican nationality and 23% (*n* = 7) were from states located in the center and south of the country (i.e., Chiapas, Oaxaca, Guanajuato). Approximately two-thirds of the sample (*n* = 21) had at least one child under the age of five, and the average age of first pregnancy for these individuals was 16.5 years old. Nearly all (93.3%) participants reported experiencing GBV in their lifetime, and a third (33.3%) reported experiencing GBV recently (past 3 months). Furthermore, 36.6% (*n* = 11) of the sample reported having ever

testing positive for an STI and 10% (*n* = 3) of the participants reported a history of substance use and one of them had been in rehabilitation. Only two participants had previously participated in a research study prior to the *Jóvenes Sanos* study.

**Table 1.** Characteristics of adolescent girls (*n* = 30) that participated in the present study in Tijuana, Mexico.

| Variable | *n* = 30 (100%) |
| :---: | :---: |
| Age (Median, IQR) | 17.5 (16–20) |
| Education | |
| Elementary (6 years) | 4 (13.3%) |
| Middle school (8 years) | 18 (60%) |
| High School (12 years) | 3 (10%) |
| Currently enrolled in school (high school) | 5 (16.7%) |
| Birthplace (birth state in Mexico) | |
| Baja California | 21 (70.0) |
| Other Mexican state | 9 (30.0) |
| Ever participated in a research project * | 2 (6.6%) |
| Ever been pregnant | 21 (70%) |
| Age of first pregnancy (Median, IQR) | 16.5 (13–18) |
| Currently using modern contraception [a] | 6 (20.0%) |
| Ever tested positive for an STI | 11 (36.6%) |
| Substance use | 3 (10%) |
| Experienced GBV | |
| Lifetime | 28 (93.3) |
| Recent | 10 (33.3%) |

* Prior to the *Jóvenes Sanos* study and/or the present study. [a] Other than condoms, including intrauterine device (IUD), Depo-Provera, implant, and/or contraceptive pills.

### 3.2. Informed Consent Vulnerabilities

Informed consent requires that an individual's decision to participate be informed, voluntary, and rational [34]. Waiving parental consent for adolescent research is a complex issue that requires a careful balance between respecting adolescent autonomy and ensuring their safety and well-being. An ethical review and oversight are critical to making informed decisions about when and how parental consent can be waived [35]. When asked about their experiences in the *Jóvenes Sanos* study, participants spoke of the importance of waiving guardian permission, the need for clear language, and the importance of individual volition to guard against the potential for coercion tied to incentives or recruitment in medical settings. Each of these specific issues will be discussed in more detail below.

'Just ask us': guardian permission as a barrier to informed consent.

Guardian permission can serve as a protective measure, but in certain research contexts, it can be a barrier to obtaining informed consent, particularly when dealing with vulnerable populations and sensitive topics. The decision to waive guardian permission in adolescent SRH research is one that poses unique ethical challenges for adolescents worldwide and is especially salient for LMICs which may lack of established ethics review procedures for determining when waiver is appropriate [36]. However, the reluctance of adolescents to participate in research when guardian permission is required is often based on fear of family stigma and punishment [37,38]. Given the cultural taboos and stigma surrounding adolescent sex in Mexico, examining the extent to which guardian permission is a barrier to SRH among LMIC girls is thus critical to research that can inform developmentally tailored SRH prevention and interventions for this vulnerable population.

When asked about the consent process, participants indicated that they appreciated they were able to consent for themselves. Some suggested that they would have felt uncomfortable or even would have opted not to participate at all were parental consent required.

*If my mom would have needed to provide consent for me, I would have felt very uncomfortable, she would have wanted to know every detail, what the study was about and I would have rather not participated...*

—Claudia, 16 years old (control site)

*I think it's best that if they just asked us [girls] if we want to participate... because what if I'm really interested and I want to participate in the study, but my parent or family member say 'no' and don't consent, then I would be left out...*

—Erika, 18 years old (intervention site)

'What's this about?': Participant misunderstanding as an obstacle for rational consent.

Although extensive efforts were made for the study participants to make an informed decision to participate in the study, some participants discussed their confusion at the outset of the consent process and did not understand what it meant to be a part of a research project.

*When they first approached me... I thought to myself 'they are going to investigate me?'... I was curious about what they wanted to know exactly about me. After she explained what she meant by a research study [estudio de investigación], the process and what the study was about, I told them I was interested in participating.*

—Angelica, 20 years old (control site)

*I felt weird at the beginning [of the study], I was a bit concerned because I didn't know what it meant to be part of a research study or what to expect... after I finished the survey, I felt more comfortable... I was nervous at the beginning because I didn't know what they were going to do to me [laughs]... I think I'm traumatized from watching all these movies about social experiments and stuff like that [laughs]...*

—Valeria, 18 years old (control site)

Thus, to adequately design developmentally appropriate consent procedures for vulnerable youths requires both assessing youths who do not understand as well as assessing their consent and comprehension abilities [34,35].

One such misunderstanding can occur when recruitment is conducted in healthcare settings as some participants reported confusion and thought the research process was part of their standard of care. Importantly, none of the participants reported feeling coerced into participating because they receive health care at that facility.

*She said this [study] was being conducted by the University of San Diego, that they were doing a survey and that they were recruited other girls as well... I found it hard to believe they wouldn't share my responses with the doctors here, but I thought it sounded interesting, so I decided to participate to learn more about what the study was about...*

—Irma, 20 years old (control site)

*The doctor [study] was very clear and told me that they wouldn't share my responses and that my participation had nothing to do with the care I received, so I told her it was fine and I agreed to participate.*

—Luisa, 16 years old (intervention site)

### 3.3. The Role of Monetary Incentives

The role of monetary incentives in motivating survey participation has been widely documented and remains an ongoing ethical debate [39]. It is essential to consider the context and objectives of a study and the target population when deciding whether to offer monetary incentives. While they can be a powerful tool for increasing participation, they should be used judiciously and in alignment with ethical principles to ensure the integrity

of the research process. Although the girls had mixed opinions, none of our participants believed the incentives for this research would induce girls to agree to a study they did not wish to participate in. Some girls reported that they did not think incentives should have been provided as they felt that the money would serve as the primary reason that other girls might choose to participate.

> *I think that it would be best if they only told them what the study was about. . . they shouldn't say that they are going to offer them money, to see if they are genuinely interested... the incentive shouldn't be mentioned because a lot of people might just do it [participate] for that reason.*
>
> —Natalia, 17 years old (control site)

Other participants felt that they benefited from the study because it was focused on their sexual and reproductive health and because they were also compensated for their time.

> *I think it was good for me [to participate] because I learned a lot about my health and also it was good to talk about it [gender-based violence] and share my opinions about it and I also received an incentive for being a part of this. . .*
>
> —Lourdes, 19 years old (intervention site)

### 3.4. A Relational Approach to Research Empowers Participants

A relational approach to research places participants at the center of the research process, emphasizing respect, collaboration, and empowerment [40]. By fostering positive relationships and valuing the contributions of participants, this approach not only produces more ethical and high-quality research but also supports the well-being and self-efficacy of those involved in the research process. Participants highlighted how the researchers' relational approach to consent and interviews helped to create a unique environment where knowledge, affirmation, and empowerment could be engendered within the girls and young women who participated in the research. Each of these themes will be discussed in detail below.

'She gave me confidence and seemed trustworthy': Importance of Relationship with Interviewer

The extent to which adolescents were able to access and benefit from participating in the research project appeared to depend on the nature of their interactions with the members of the research team. Research team members play a critical role in establishing and maintaining a rapport with study participants. This practice not only enhances the quality and reliability of research data but also upholds ethical principles and ensures a positive and respectful experience for participants [40]. Consistent with previous work in other LMICs involving medically underserved women and girls, participants greatly valued their relationships with research staff whom they perceived as trustworthy and caring [41], which impacted their decision to divulge sensitive information or agreeing to participate in research.

> *She [study staff] invited me to participate in the study and made me feel comfortable. . . she gave me confidence and seemed trustworthy. . . she was very kind and explained to me all the procedures and she told me that if I wanted to stop the interview at any time it was okay, to just let her know. . .*
>
> Erika, 16 years old, (intervention site)

'Topics that are hard to talk about': De-Stigmatizing Discussion of SRH.

De-stigmatizing SRH topics in research with adolescent girls in LMICs is essential to ensure that their unique needs and experiences are effectively addressed. Stigma can hinder open communication, access to care, and the success of research initiatives [42]. The majority of the participants (90%) reported they thought that it was useful to include SRH themes in research and they gained important SRH knowledge by participating in the study.

*I actually learned a lot including how to be more confident and trust myself. . . it's important to talk more about these topics that are hard to talk about, for example, HIV. We know it exists because we get tested here, but they don't talk about how to prevent it and how to take care if you get it. . .so I think doctors should talk about that during periodic visits that we have here. . .*

—Alicia, 19 years old (control site)

*I think that it's very useful to continue doing this type of research because there are a lot of adolescents that are not well informed, and they don't know 100% what STIs are or how to prevent them even though they are sexually active. I think that thanks to this research and the topics they covered, they are educating us, and we are learning a little bit more than we knew before.*

—Sofia, 18 years old (intervention site)

### 3.5. Destigmatizing Gender-Based Violence (GBV)

Research on GBV among adolescents in LMICs plays a crucial role in informing policies, programs, and interventions aimed at preventing and addressing this issue. By understanding the dynamics of GBV, identifying effective strategies, and advocating for change, researchers can contribute to the well-being and safety of adolescents in these regions. However, one common ethical issue often raised in trauma-related research is whether the inclusion of sensitive questions heightens psychological distress. Previous research suggests that, while reactions to questions of abuse are variable, almost all individuals see the benefits of the research for themselves and others, and report that they would be willing to participate in similar research again [43]. Responses from participants in the current study highlight this same idea.

*. . . She [interviewer] talked about violence against women, and I think that's good. . . questions about if I would let someone hurt me. . . or what would I do if someone hurt me. . . and it makes me think of some experiences that I had in the past and I feel more aware now. . . those experience happen to a lot of adolescents here in our community. . .*

—Sandra, 16 years old (control site)

*I liked talking about domestic violence [gender-based violence] because many women do not know that they are experiencing sexual violence from their partners, so then, I imagine that they do it [have sex] to please their husbands but they don't realize the harm that it can do, right? I think if they don't feel comfortable, they shouldn't do it. . .*

—Guadalupe, 18 years old (intervention site)

### 3.6. Sense of Empowerment

The majority of the participants reported they felt that their involvement in the study was beneficial or useful for them. Participants discussed how their individual relationship with research personnel allowed for a sense of empowerment, including the initiative to speak to their healthcare provider about family planning.

*Before [participating in] this study, my mom would be with me all the time during my visits, and she would respond to the doctors for me. I did not feel safe or comfortable when responding to them by myself. . . my mom had to come with me everywhere but when I was invited to participate, I felt safe with the interviewer and after this, I've felt more confident and going to my visits alone now. It's something that I am proud of.*

—Maria, 16 years old (intervention site)

*She taught me about the methods to take care of yourself [birth control], I only knew about the injections and the device, the IUD (intrauterine device), and she told me about other options that I didn't know about. . . and I liked learning about it and making a more informed decision. . . in the end I decided to get the device [IDU].*

—Priscilla, 16 years old (intervention site)

## 4. Discussion

In this study, we analyzed the perceived risks and benefits of participation in SRH research among adolescent girls in Mexico. Responses centered around two primary themes: the importance of ensuring that consent to participate is tailored to the needs of adolescent participants and the importance of the researcher–participant relationship in facilitating participant empowerment.

Our results suggest that adolescent girls derive satisfaction from their ability to independently understand and make decisions regarding their decision to participate in SRH research without the involvement of their guardians, aligning with previous research that has suggested that parental consent is a barrier in SRH research [44–46]. If the adolescents are deemed mature enough to understand the research and make independent, informed decisions about their participation, their autonomy may be respected. The assessment of maturity and capacity is critical in determining whether adolescents can provide valid and voluntary consent [47]. However, it is also important to acknowledge that the legal framework regarding consent for minors varies by country and jurisdiction. The decision to waive parental consent is context-specific, and careful ethical considerations must be applied in each case. For example, some LMICs, such as South Africa, require researchers to obtain consent from those under 18 years of age as well from a guardian before they can participate in research [48]. Limiting research enrollment to adolescents who can obtain guardian consent may lead to non-representative findings and a continued lack of developmentally appropriate interventions [49]. In these settings, researchers should explore alternative mechanisms to ensure the protection of participants, such as seeking assent from the adolescents, providing information to parents without requiring formal consent, or involving trusted adults in the process (e.g., "*in loco parentis*") [47,50].

At the same time, power imbalances [32] between an investigator and an adolescent participant and the subsequent increased risk for vulnerability require special attention when guardian consent is waived. Therefore, careful attention must be paid to ensure that the consent process respects the autonomy, rights, and welfare of adolescents [34]. Engaging with the community and relevant stakeholders, including adolescents themselves, can provide valuable insights into the acceptability and appropriateness of waiving parental consent [51,52]. Community input and the involvement of community advisory boards can help ensure that the research is culturally sensitive and ethically sound.

The results from this study also highlight the consent vulnerabilities that emerge when research is conducted in healthcare settings. In these contexts, the power dynamic between healthcare providers and patients can create a situation in which patients may feel compelled to participate in research out of a sense of obligation to their healthcare provider, potentially compromising the voluntary nature of consent [53,54]. Further, patients in healthcare settings may misconstrue the research study as part of their medical treatment rather than a distinct research endeavor. This misunderstanding can also affect the voluntariness of their participation [55]. Likewise, an invitation to participate in a research study may lead to confusion about the investigator's role and fear that failure to consent will result in denial or discontinuation of services, highlighting the critical need for researchers to ensure full comprehension and the absence of coercion, including a thoughtful approach pertaining to participant compensation. Researchers must implement robust strategies to ensure the ethical conduct of research in health care settings. This includes providing clear and understandable information, allowing sufficient time for consideration, obtaining voluntary and informed consent, and considering additional safeguards for vulnerable populations [56–58].

Furthermore, our interviews underscore how perceptions of comfort and safety within the researcher–participant relationship were critical to the success of the parent study. A trusting relationship allowed participants to explore their views on HIV and STIs, topics viewed as taboo for many in their culture, and of which they had previously been uncomfortable discussing in health settings [59]. Adolescent girls who participated in this study reported a sense of empowerment as a result of the information they learned and their

relationship with research staff. Further, several participants indicated that discussing GBV in a destigmatizing environment was valuable as many of them had experienced GBV but had not previously disclosed this to others. Contrary to concerns often expressed by ethics review committees, discussions with staff regarding STIs and GBV did not elicit or 'trigger' traumatic responses in this sample. Additionally, participants benefited from a safe space where they could openly discuss their concerns, questions, and experiences. They were also referred to health and social services for victims of GBV that were free or low cost.

*Limitations*

The findings of this study must be considered alongside their limitations. First, the results of this study were drawn from a small sample of participants aged 16–20 who opted to participate in the *Jovenes Sanos* study 3 months after completing their participation. Thus, findings may not be generalizable to younger adolescents, be representative of those who did not participate in the parent study and may create a risk of recall bias. Quantitative research with representative samples of adolescent girls from such contexts is needed to understand their experiences as study participants in research focused on SRH and to improve SRH interventions. Nevertheless, considering the limited research on adolescents' experiences as study participants in SRH research in Mexico, this study provides important insights into these ethical issues and areas for future research. Furthermore, although this study took place prior to the COVID-19 pandemic, we find that the results are still relevant, given the increase in GBV, STIs, and adolescent pregnancy experienced in Mexico and across other Latin American countries, further highlighting the need for research focused on improving the SRH of adolescent girls in LMICs.

## 5. Conclusions

There are significant unmet sexual and reproductive health (SRH) needs in adolescents in LMICs. Limitations notwithstanding, our results provide important information for conducting ethical research in this domain. Our findings suggest that researchers interested in involving adolescent girls in research related to SRH should respect both the importance of the autonomy and dignity of participants as developing persons and the significance of a trusting relationship with study personnel. Paying attention to such concerns throughout the consent process and throughout the study itself creates conditions in which participants can feel a sense of empowerment in raising SRH issues with healthcare providers and allows for a higher quality of data only possible through the establishment of trusting relationships. Tailoring sexual health research practices to meet the needs of adolescent girls in LMICs requires a comprehensive, culturally sensitive, and participant-centered approach. This approach ensures that research is respectful, ethical, and effective in addressing the unique challenges faced by adolescent girls in these regions.

**Author Contributions:** A.E.S. conducted in-depth interviews, supervised transcription efforts, lead coding and data analysis, contributed to interpreting results and to writing the manuscript; R.M. helped develop interview guides and contributed to writing the manuscript; S.W. helped interpret the data/results and contributed to writing the manuscript; T.R.-J. conducted in-depth interviews, supported transcriptions, coding and interpreting results under the guidance of A.E.S. and C.B.F.; S.E.O. supported data analysis, interpreting results under the guidance of A.E.S. and C.B.F. and preparing manuscript for submission.; G.M.R. supported data collection efforts and contributed to writing the manuscript; C.B.F. helped develop interview guides, lead coding and data analysis, contributed to interpreting results and writing the manuscript. All authors have read and agreed to the published version of the manuscript.

**Funding:** This work was supported by the Eunice Kennedy Shriver National Institute on Child Health and Human Development (NICHD) (K23HD084756, PI: Servin) and the Fordham University HIV and Drug Abuse Prevention Research Ethics Training Institute (RETI) and the National Institutes on Drug Abuse (NIDA) (R25 DA 031608; PI: Fisher).

**Institutional Review Board Statement:** The study was conducted according to the guidelines of the Declaration of Helsinki and approved by the Human Research Protections Program at the University of California San Diego (protocol #181877S) and the Institutional Review Board (IRB) of the School of Medicine, at the Universidad Xochicalco, in Tijuana, Baja California, Mexico.

**Informed Consent Statement:** All participants provided written informed consent prior to participating in this study.

**Data Availability Statement:** The datasets used and/or analyzed during the current study are available from the corresponding author on reasonable request.

**Acknowledgments:** The authors would like to extend a special gratitude to the adolescent girls who participated in this study. The authors also gratefully acknowledge the entire multidisciplinary research team involved in this project, the adolescent girls from the Youth Advisory Board (YAB), the Mexican Ministry of Health (ISESALUD), and Alejandra Padilla Mercado for their assistance with data collection and dissemination of study findings.

**Conflicts of Interest:** The authors have no potential competing interests with respect to the research, authorship, and/or publication of this article.

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
