# Peer review of "Tailoring Sexual Health Research Practices to Meet the Needs of Adolescent Girls in Low- and Middle-Income Countries: Findings from Mexico"

_adolescents, doi:10.3390/adolescents4010011_

Round 1
Reviewer 1 Report
Comments and Suggestions for Authors
This article investigates the perceived risks and benefits of participating in SRH research (of adolescents in Mexico)--focusing particularly on the ethical considerations underpinning SRH research with adolescents--in order to not only improve the quality of SRH research with adolescents but, moreover, to further the design and implementation of effective interventions targeting the SRH of adolescents in a context characterized by GBV (especially IPV), and teenage pregnancy.
The paper is clearly written and structured. The introduction tells the reader exactly what to expect, and the paper unfolds as expected.
The authors highlight, first, the importance of ensuring that consent is developmentally tailored to the needs of participants and, second, the role that trust and a sense of safety in the relationship between researcher and participant plays in the quality and ethical unfolding of this research. In regards to the latter, they also indicate the importance of the researcher destigmatising discussions of SRH and GBV, and working towards offering participants a sense of empowerment.
I thought the discussion of the parental waiver could have been a bit more thorough, and wonder whether the authors want to engage with the assumption that adolescents are immediately, inherently, or necessarily vulnerable (both in general, and when it comes to SRH research in particular).
See small comments in the pdf attached.

Author Response
The authors would like to thank the reviewer for their time and consideration. Please see the attachment.

Reviewer 2 Report
Comments and Suggestions for Authors
First, I would like to thank the editor for the opportunity to review this paper. The work may be of relevance to the scientific community. In my opinion, with some minor improvements it could be more valid for publication.
- - Introduction and Discussion. The theoretical foundation seems insufficient to me. The importance of the work presented should be expanded. Likewise, I believe that the discussion is also a bit thin and that the contrast should be expanded with more abundant scientific literature.
- - Methods.
o It is not well understood how the sample goes from 100 participants to 30.
Comments on the Quality of English LanguageMinor editing of English language required.
Author Response

(The authors gave the same response as above.)

Reviewer 3 Report
Comments and Suggestions for Authors
Thank you for inviting this reviewer to review this article entitled, " Tailoring sexual health research practices to meet the needs of adolescent girls in low-and-middle income countries: findings from Mexico" being submitted to Adolescents journal.
Abstract: no comments
Introduction
The reviewer suggests that the authors consider a broader statement on adolescent pregnancy or potentially start with statistics. The first sentence (see line 43) had the reviewer questioning who was “these” adolescents. As if the reader would know exactly what was meant.
The reviewer suggest that the authors consider adding a few lines about the Youth advisory Board in the introductionand its role and importance in this type of research.
The reviewer suggests authors consider adding references among Lines 59-73 in order to justify some of the broader statements.
The reviewer suggest that the authors consider mentioning and explaining the previous study prior to the end of the introduction section ( see lines 69-73). The reviewer suggest that the authors start a new paragraph in the introduction that speaks to the original study, what it was, is there a reference to the previous study.
Materials and Methods
It is unclear to the reviewer at this point in the manuscript how the sub-study participants were identified if the parent study was a randomized control trial pilot. How were the adolescent girls from the Jovenes Sanos intervention identified, recruitment method should be named, snowball, convenience, etc… (see lines 75-82).
It was unclear to the reviewer whether the initial study was a quantitative study, whether a survey was used.
Line 96 insert “was” before composed.
Line 97-98, would take out striving to vocalize…… and add to piece in the introduction.
The reviewer suggest that the authors add headings such as Sample population and recruitment under the materials and methods section to make this clear.
Line 106 is where the reviewer first understands what type of study the previous study was, the reviewer suggest that information come much earlier in introduction and perhaps earlier in materials and methods and not in data collection.
The reviewer suggest that the authors provide a bit more information on how they reached thematic saturation and provide a reference for their method. Was the bootstrapping method used?
Results
Line 137-says GBV “recently” the reviewer suggest that the authors define what recent means and the parameters, last month, three months, or a year?
This reviewer suggest that table 1 be aligned so that there is a column for % and n which would make the table easier to read. The reviewer also suggest that perhaps two tables could be made one with the traditional demographic descriptors and the other relating to the participants sexual and reproductive characteristics.
The reviewer agrees that comprehension and age are intertwine with literacy, which plays a role in whether true informed consent can be obtained and the reviewer wonders if it would be possible to flesh this out in the IDI’s and well as the influence of the media. (see lines 183-193).
See line 255- a larer space than usual detected here for some reason.
Discussion
The reviewer suggests that the topic of power imbalance and dynamics might come earlier than in the discussion portion, the authors might also consider it in the limitations as well as a more in-depth discussion around comprehension, literacy, and age.
Conclusions
The reviewer suggest that the authors might add to conclusions by weritign about future direction as well.
Author Response
The authors would like to thank the reviewer for the time and consideration taken to review this study. Please see the attachment.

Reviewer 4 Report
Comments and Suggestions for Authors
It is quite an interesting article focusing upon tailoring sexual health research practices to meet the needs of adolescent girls in LMICs.
The findings of this research are from Mexico, and this could be taken as a preliminary study focusing on the low-and-middle-income countries of South America, since the most articles concerning the same topic are referred to LIMCs of Africa and Asia, and according to the references there is only one article from Mexico (number 5).
There are some aspects that should be consider.
· In the abstract the scope of the research isn’t clearly defined and should be added that tailoring sexual health research practices to meet the needs of adolescent girls in LMICs requires a comprehensive, culturally sensitive, and participant-centered approach.
· It isn’t necessary to write the discussion of the adolescent girls with the research team. On the contrary, it could be better for the reading of the article, the answers of the participants to be integrated as a text.
· Also, the names of the girls should be deleted and not mentioned into the text.
· The number 21 reference should be deleted since it refers to males interested in sex with males and is irrelevant with the article title and theme.
The number of participants is small and can’t be representative
Author Response
The authors would like to thank the reviewer for their time and consideration in reviewing this study. Please see the attachment.

Reviewer 5 Report
Comments and Suggestions for Authors
I appreciated reading this paper and found it to be interesting and generally well-written. However, the introduction needs to better motivate the paper. While the end of the introduction states, “to bridge knowledge gaps in ethical challenges that have not been considered or addressed in past research efforts in Mexico.”, however this gap was not demonstrated.
I am not sure what this sentence is trying to say, “A Youth Advisory Board (YAB) established by the Mexican Ministry of Health (Secretaria de Salud del Estado de Baja California) composed of 50 adolescent girls (ages 15–18) from the school districts neighboring the community health centers striving to vocalize their needs by engaging in advocacy and other activities.”
How was the interview guide developed? Tested? Deemed appropriate to answer the research question?
The theme titled, “Incentives shouldn’t be mentioned” seems misnamed. Two quotes were presented, and only one of them mentioned that. The other said she was glad to have the incentive.
The discussion section could use some deepening. In what ways does this investigation move the conversation forward? As currently written, the discussion makes it seem that this current paper only confirms what has already been written.
Comments on the Quality of English LanguageJust the one problem I noted above.
Author Response

(The authors gave the same response as above.)
